# Regulation by the quorum sensor from Vibrio indicates a receptor function for the membrane anchors of adenylate cyclases

Stephanie Beltz[1], Jens Bassler[2], Joachim E Schultz[1]*

[1]Pharmazeutisches Institut der Universität Tübingen, Tübingen, Germany; [2]Max-Planck-Institut für Entwicklungsbiologie, Tübingen, Germany

**Abstract** Adenylate cyclases convert intra- and extracellular stimuli into a second messenger cAMP signal. Many bacterial and most eukaryotic ACs possess membrane anchors with six transmembrane spans. We replaced the anchor of the AC Rv1625c by the quorum-sensing receptor from *Vibrio harveyi* which has an identical 6TM design and obtained an active, membrane-anchored AC. We show that a canonical class III AC is ligand-regulated *in vitro* and *in vivo*. At 10 μM, the cholera-autoinducer CAI-1 stimulates activity 4.8-fold. A sequence based clustering of membrane domains of class III ACs and quorum-sensing receptors established six groups of potential structural and functional similarities. The data support the notion that 6TM AC membrane domains may operate as receptors which directly regulate AC activity as opposed and in addition to the indirect regulation by GPCRs in eukaryotic congeners. This adds a completely novel dimension of potential AC regulation in bacteria and vertebrates.

*For correspondence: joachim.
schultz@uni-tuebingen.de

## Introduction

In 1958 Sutherland and Rall reported the structure of a second messenger, cyclic 3′,5′-adenosine monophosphate (cAMP) which was generated upon incubation of a liver extract with the first messengers epinephrine or glucagon (*Sutherland and Rall, 1958*). Since then cAMP has been demonstrated to be a universal second messenger translating a variety of extracellular stimuli into a uniform intracellular chemical signal. The enzymes responsible for biosynthesis of cAMP from ATP, adenylate cyclases (ACs), have been biochemically and genetically identified in most bacterial and eukaryotic cells (*Khandelwal and Hamilton, 1971*; *Linder and Schultz, 2003*; *2008*). To date, sequencing has identified six classes of ACs. The small-sized class I (enterobacterial ACs), class II (toxin class) and the minor classes IV-VI are restricted to bacteria (*Bârzu and Danchin, 1994*; *Linder and Schultz, 2003*). The class III ACs are ubiquitous, albeit with differing domain architectures (*Linder and Schultz, 2008*). The catalytic domains share sequence and structural similarities, yet minor, characteristic sequence peculiarities have resulted in a division into four subclasses, a-d (*Linder and Schultz, 2008*; *Tesmer et al., 1997*; *Tews et al., 2005*). A fundamental difference between bacterial and eukaryotic ACs is that the former are monomers which require homodimerization for activity. The mammalian congeners, exclusively class IIIa, present themselves as pseudoheterodimers composed of two concatenated 'bacterial' monomers with slightly diverged, yet complementary domains (*Guo et al., 2001*). Accordingly, mammalian class III ACs are anchored to the membrane by two putative 6TM bundles, one in each of the concatenated repeats. Our knowledge about regulation of bacterial class III ACs is limited. Apart from a few soluble ACs which appear to be regulated by carbon dioxide or pH near to nothing is known (*Kleinboelting et al.,*

**eLife digest** Cells are surrounded by a membrane that separates the inside of the cell from the external environment. To communicate information across the cell membrane, cells often employ a relay system. In this system, receptor proteins on the surface of the cells sense information about the environment and trigger the production of a chemical signal inside the cell.

Certain receptors activate enzymes called adenylate cyclases, which reside just inside the cell, to produce a chemical signal. In some human and bacterial adenylate cyclases, about 40% of the protein is anchored in the membrane, far more than is necessary to hold the protein in place. It is therefore possible that this "membrane anchor" region plays an additional role, perhaps even detecting external signals.

A "quorum sensing" receptor protein that was recently discovered embedded in the membrane of a species of marine bacteria called *Vibrio harveyi* has a similar structure to the membrane anchor of adenylate cyclases. Beltz et al. have now replaced the adenylate cyclase membrane anchor with a *V. harveyi* receptor. This produced a hybrid protein that could both receive and translate signals from the membrane receptor.

A computational analysis of the membrane anchors of adenylate cyclases showed that they have striking similarities to quorum sensors. Furthermore, the membrane anchors of different types of adenylate cyclase have diverse structures that may have helped the cyclases to adapt to different environments and biological requirements.

Overall, Beltz et al.'s results suggest the adenylate cyclase membrane anchor is a new type of cell surface receptor. In the future it will be important to identify the environmental signal that activates adenylate cyclases, both in bacteria and mammals.

*2014*; *Steegborn et al., 2005*; *Tews et al., 2005*). The established regulation of the nine mammalian, membrane-delimited AC isoforms is indirect. Stimulation of G-protein-coupled receptors (GPCRs) by extracellular ligands releases Gsα intracellularly which binds to ACs and activates. A potentially direct ligand-regulation of class III ACs via their large membrane anchors remains a genuine possibility.

The 6TM membrane anchors of bacterial ACs are obviously structural analogs of the 6TM bundles in mammalian ACs. In the past, we replaced the membrane anchor of the mycobacterial AC Rv3645 by the *E. coli* chemotaxis receptors for serine, Tsr, and aspartate, Tar (*Kanchan et al., 2010*). Tsr/Tar and Rv3645 have a signal-transducing HAMP domain in common and both require dimerization. The chimeras were regulated *in vitro* and *in vivo* by serine or aspartate (*Kanchan et al., 2010*; *Mondéjar et al., 2012*; *Winkler et al., 2012*), i.e. a 2TM receptor, Tsr or Tar, with an extensive periplasmic ligand-binding domain replaced a 6TM AC membrane anchor which lacks periplasmic loops. The data demonstrated that in principle direct regulation of a class III AC via an extracellular ligand is a possibility. The question whether also a 6TM receptor might directly regulate a 6TM AC remained open. This question is addressed here.

Membrane anchors with 6TMs are present in many proteins. Often they have short transmembrane-spanning α-helices and short connecting loops, e.g. in bacterial and mammalian ACs (*Krupinski et al., 1989*; *Linder and Schultz, 2003*), in the cytochrome subunits of succinate dehydrogenases and fumarate reductases (*Hederstedt, 1998*; *Yankovskaya et al., 2003*), ABC transporters (*Chang and Roth, 2001*), in bacterial HdeD proteins (*Mates et al., 2007*), **s**ix **t**ransmembrane **e**pithelial **a**ntigen of the **p**rostate (STEAP, *Kleven et al., 2015*), or quorum-sensing (QS) receptors from *Vibrio* and *Legionella* which have His-kinases as cytosolic effectors (*Ng and Bassler, 2009*). For the latter lipophilic ligands have been identified (*Ng et al., 2011*; *2010*). This has opened the opportunity to replace the 6TM anchor of the mycobacterial class IIIa AC Rv1625c which is considered to be an ancestral form of mammalian ACs (*Guo et al., 2001*), by the prototypically identical 6TM QS-receptor CqsS from *V. harveyi* (*Figure 1*). Here, as a proof of principle, we demonstrate that a 6TM receptor not only substitutes a membrane-anchoring function, but also confers direct regulation of a class IIIa AC via an extracellular ligand with nanomolar potency. Taken together with a bioinformatic analysis the data indicate that the 6TM AC membrane anchors

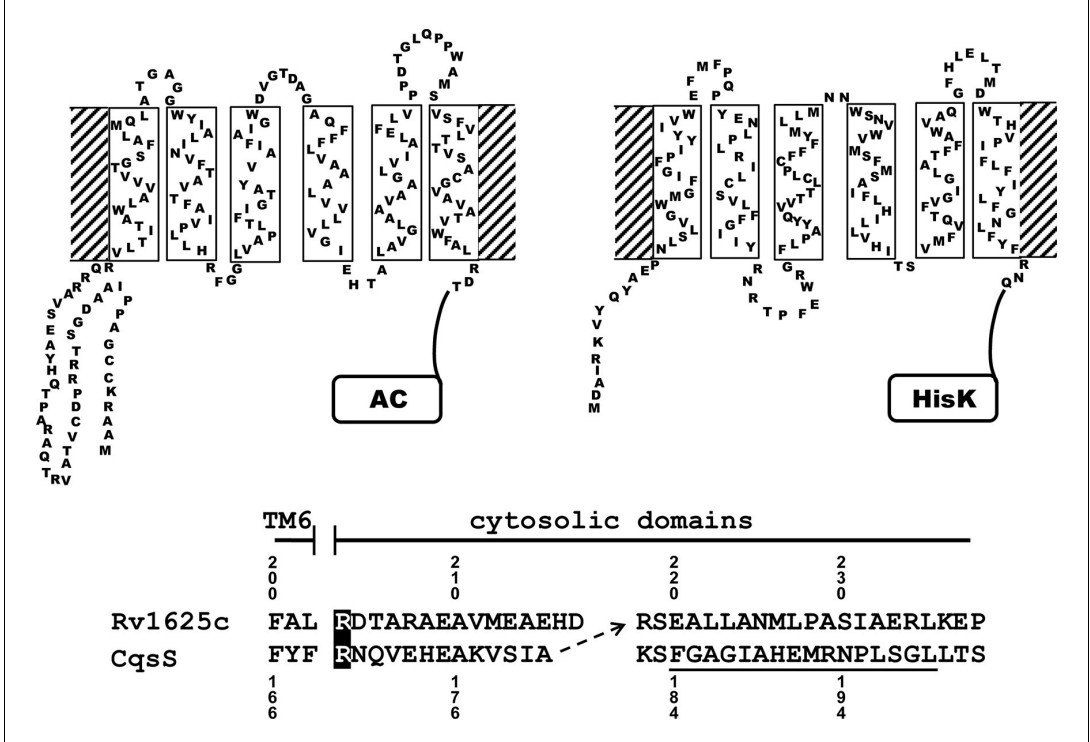

**Figure 1.** Two-dimensional models of the canonical class IIIa adenylate cyclase Rv1625c from *M. tuberculosis* (left) and the quorum-sensing receptor from *V. harveyi* (right) in the membrane. Both proteins require dimerization to be catalytically active. The alignment below covers the amino acid sequences at the exit of TM6 of both proteins. The most efficient functional linkage of the CqsS receptor to the catalytic domain of Rv1625c is indicated by an arrow. The H-Box of the histidine-kinase domain is underlined. The numbering for CqsS and Rv1625c is indicated above and below the respective sequences.

probably have an additional receptor function. This is a first and decisive step to add a novel dimension of direct regulation of class IIIa AC activities.

## Results

### The quorum-sensing receptor CqsS from *V. harveyi* regulates the class IIIa AC Rv1625c

Many bacterial and the nine membrane-bound mammalian class IIIa ACs possess 6TM modules as membrane anchors (*Guo et al., 2001*; *Linder and Schultz, 2003*; *2008*). They comprise >40% of the proteins. A function of these membrane domains beyond fixation in the membrane is unknown. Recently we became aware of the QS-receptors from *Vibrio* and *Legionella*, CqsS and LqsS, which feature a membrane anchor of an essentially identical design as the aforementioned ACs, i.e. minimal-length α-helices and short connecting linkers which presumably severely restrict conformational possibilities (see *Figure 1* for a 2D representation; *Ng et al., 2010*; *Tiaden and Hilbi, 2012*). For both QS-receptors highly lipophilic ligands have been identified such as CAI-1, the **C**holerae **A**uto**In**ducer-**1**, (S)-3-hydroxy-tridecan-4-one and LAI-1, the **L**egionella **A**uto**In**ducer-**1**, (S)-3-hydroxy-penta-decan-4-one (*Ng et al., 2010*; *Spirig et al., 2008*). The QS-receptors from *Vibrio* and *Legionella* are homodimers linked to histidine-kinases as cytosolic effector domains (*Ng and Bassler, 2009*). This is comparable to bacterial class III ACs which are homodimers (*Linder and Schultz, 2003*). These superficial observations suggested that by swapping the membrane anchor/receptor between a 6TM AC and CqsS from *Vibrio* one might generate a 6TM AC which is regulated by the QS-ligand CAI-1. We chose the mycobacterial class IIIa AC Rv1625c for this investigation because of its similarity to the mammalian congeners (*Guo et al., 2001*) and the QS-receptor from *V. harveyi*.

The success of generating a functionally productive chimera between the AC and the QS-receptor hinges on the precondition that a suitable point of transition between both proteins can be found which allows signal propagation from the CqsS receptor to the AC effector. The cytosolic aa sequences exiting from the respective last TMs have no conspicuous complementarity which would indicate a self-evident point of connection (*Figure 1*). Therefore, 15 different points of connection between the CqsS receptor and the catalytic domain of Rv1625c were probed. The first points of transition tested were the arginine residues present at the cytosolic membrane exit in both proteins (*Figure 1*). The chimera $CqsS_{1-168}Rv1625c_{203-443}$ was active (16.2 nmol cAMP·mg$^{-1}$·min$^{-1}$), but unregulated. The basal AC activity of this chimera was comparable to that of the membrane-bound Rv1625c holoenzyme (*Guo et al., 2001*), i.e. generally the two disparate domains were functionally fully compatible with each other. In the cyanobacterial class IIIa AC CyaG from *Arthrospira platensis* a distinct N-terminal domain that starts with RSEELL, was required for a functional interaction with the chemotaxis receptor Tsr (*Winkler et al., 2012*). A comparison between CyaG and Rv1625c AC sequences revealed a similar domain in Rv1625c beginning with RSEALL (*Figure 1*). Hence, for the Rv1625c AC this point of transition to CqsS was chosen. In the CqsS His-kinase the auto-phosphorylated histidine is part of the canonical H-box (underlined in *Figure 1*; *Grebe and Stock, 1999*). However, several chimeras of CqsS and Rv1625c linked in this region were unaffected by CAI-1. With transition points closer to the membrane exit of the CqsS receptor, e.g. at Val172, Ala181 or Gly185, AC activities were reproducibly stimulated by CAI-1 (not shown). For further experiments we linked Ala181 of the QS-receptor to Arg218 of the Rv1625c AC, generating $CqsS_{1-181}$-$Rv1625c_{218-443}$ (abbreviated CqsS-Rv1625c; *Figure 1*) because it responded maximally. The chimera CqsS-Rv1625c was stimulated by 85% with 10 μM CAI-1 (*Figure 2*). The response was concentration-dependent and the EC$_{50}$

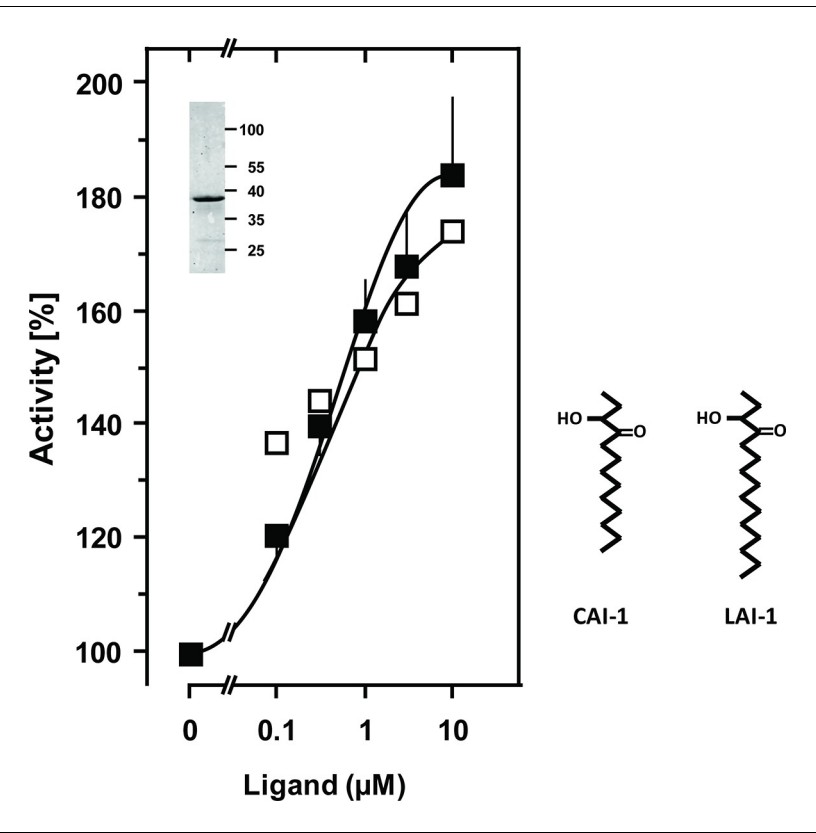

**Figure 2.** Stimulation of the chimera $CqsS_{1-181}$-$Rv1625c_{218-443}$ by the QS-ligands CAI-1 or LAI-1. Basal activity was 5.5 nmol cAMP·mg$^{-1}$·min$^{-1}$. The EC$_{50}$ concentrations were 400 nM. Filled squares, CAI-1 (n = 5–12; ± S.E.M.); open squares, LAI-1 (n= 1–2). CAI-1 stimulations were significant starting at 100 nM ligand. The insert shows a Western blot of the expression product with MW standards indicated at the side. The structure of the ligands is depicted at right. The catalytic domain of Rv1625c alone was not affected by CAI-1 or LAI-1 (not shown).

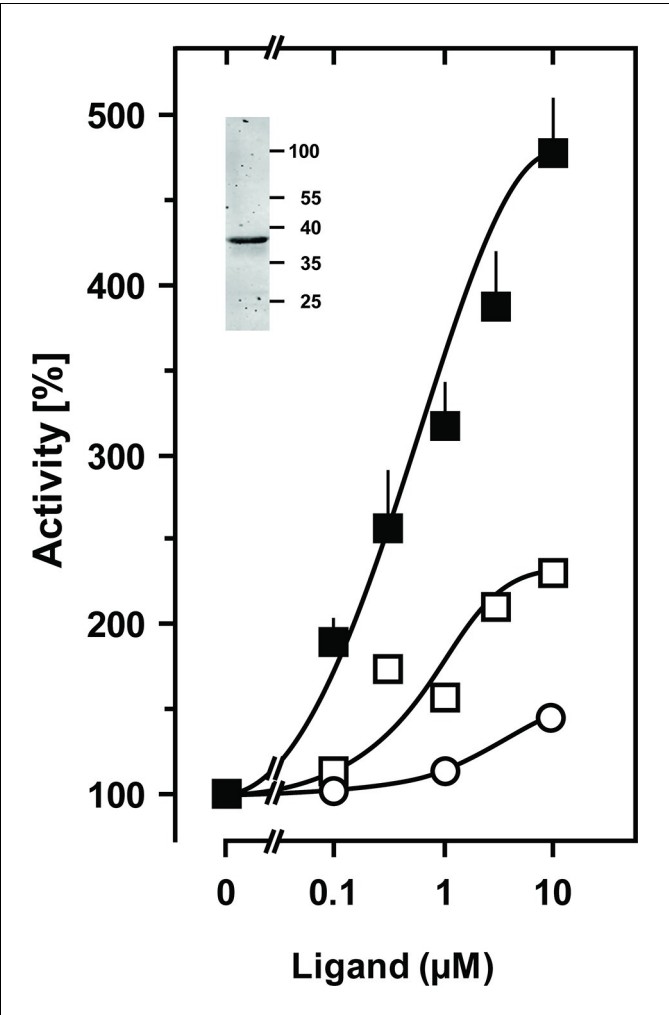

**Figure 3.** Stimulation of the chimera CqsS$_{1-181}$F166L-Rv1625c$_{218-443}$ by the QS-ligands CAI-1 or LAI-1. Basal activity was 4 nmol cAMP·mg$^{-1}$·min$^{-1}$. Filled squares, CAI-1 (n = 5–11; ± S.E.M.); open squares, LAI-1 (n=2); open circles, 3,4-tridecanediol. The EC$_{50}$ concentrations were 400 nM CAI-1, 900 nM LAI-1, and 2000 nM 3,4-tridecanediol. CAI-1 stimulations were significant starting at 100 nM ligand. Insert: Western blot of expression product.

concentration for CAI-1 was 400 nM (*Figure 2*). Discrimination between CAI-1 with a C9 and LAI-1 with a C11 lipid tail was absent. In both cases maximal stimulation at 10 µM CAI-1 and the EC$_{50}$ concentrations were identical (*Figure 2*). These parameters perfectly matched the concentrations of 400 nM required to observe phenotypic responses from the CqsS-His-kinase in *V. cholerae* (*Ng et al., 2011*).

*In vivo* characterization of CqsS from *V. cholerae* identified Cys170 at the exit of TM6 as critical for signaling (*Ng et al., 2011*). In CqsS from *V. harveyi* the corresponding residue is Phe166 (*Figure 1*). We examined its role by substituting it with all 19 proteinogenic amino acids. Substitutions by large flexible hydrophobic residues, i.e. Leu, Ile, or Met enhanced CAI-1 stimulation up to 480% compared to 185% with the parent Phe166 construct (compare *Figures 2* and *3*).

Furthermore, a high discrimination between CAI-1 and LAI-1 was effectuated and the potency ratio CAI-1/LAI-1 at 10 µM increased from 1 to 3 (see *Figures 2* and *3*). Evidently, the F166L CqsS sensor domain in the chimera shows high selectivity between CAI-1 and LAI-1. Our findings were in line with earlier results that the natural CqsS receptor preferably detects ligands with C$_{10}$ or C$_{8}$ but not shorter (C$_{6}$, C$_{4}$) or longer (C$_{12}$, e.g. LAI-1) alkyl tails (*Tiaden and Hilbi, 2012*). 3, 4-tridecanediol which has not been reported as a natural ligand stimulated by about 10% of CAI-1 at 10 µM (*Figure 3*). Other amino acid substitutions at position 166 of CqsS did not affect the extent of AC

stimulation with the exception of tryptophan which abrogated stimulation (data not shown). Activation enhanced Vmax from 14 to 42 nmol cAMP·mg$^{-1}$·min$^{-1}$ whereas Km for substrate ATP was not significantly affected (233 and 163 µM ATP, respectively). An up to 10$^4$-fold activation of CqsS-stimulated reporter gene transcription was earlier observed in *V. harveyi* and *V. cholerae* (21). The 5-fold activation of CqsS-F166L reported here appears comparatively small. This discrepancy can be explained by the fact that the amplification systems used differ profoundly. Earlier studies investigated the quorum-sensing system *in vivo* with a reporter gene transcription/bioluminescence readout which is much more sensitive than an *in vitro* AC assay with isolated cell membranes used in the present study (*Ng et al., 2011*). In addition, the coupling of CqsS to its authentic effector might well be more stringent than that attainable in a chimera with an exogenous Rv1625c AC output domain.

Next we examined whether the CqsS-Rv1625c AC chimera is operational *in vivo*. Use of maltose by *E. coli* requires activation of the maltose operon via the cAMP/CRP signaling system. Maltose fermentation produces organic acids which are visualized on MacConkey plates by the pH indicator phenol red. We used the AC-deficient *E. coli cya-99* strain with a high affinity CRP variant (*Garges and Adhya, 1985*). It cannot metabolize carbohydrates for lack of cAMP. When grown on MacConkey agar, colonies appear whitish. We transformed the CqsS-F166L-Rv1625c into *E. coli cya-99*, plated the cells on MacConkey maltose agar and induced AC expression by a filter strip soaked with 30 mM IPTG. The reddish zone along the filter strip which is indicative of maltose metabolism was expanded at the side where 10 µl of a 100 µM CAI-1 solution was applied (*Figure 4*), clearly demonstrating a CAI-1 stimulated cAMP production *in vivo*.

## CqsS dimerization is required for adenylate cyclase regulation

The monomeric bacterial class III ACs require homodimerization (*Linder and Schultz, 2003*). Similarly, bacterial His-kinases of two-component systems are homodimers in which a His-residue of the H-box is phosphorylated either in cis or trans (*Casino et al., 2014*; *2009*). Here we examined whether dimerization of the QS-receptors is required for AC regulation. The CqsS receptor was connected to known inactive Rv1625c AC point mutants, Rv1625cD300A and Rv1625cR376A. These point mutants complement each other and the dimer is catalytically active (*Guo et al., 2001*). Thus, CqsS-Rv1625cD300A was inserted into pETDuet-3 alone or in combination with CqsS-Rv1625cR376A. When CqsS-Rv1625cD300A and CqsS-Rv1625cR376A were jointly expressed, robust

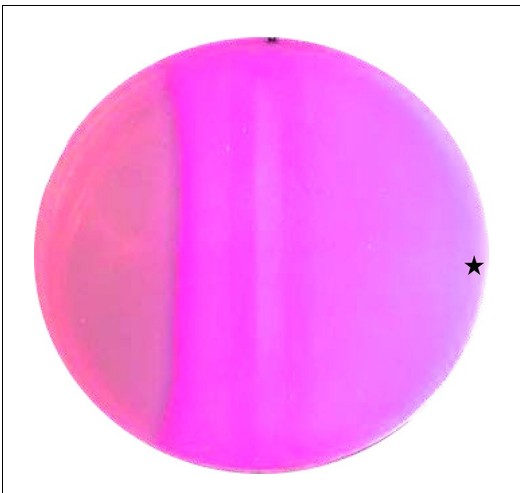

**Figure 4.** CAI-1 stimulates cAMP formation *in vivo*. A MacConkey maltose agar plate with *E. coli cya-99 crp*144* transformed with CqsS$_{1-181}$F166L-Rv1625c$_{218-443}$ was induced by a filter strip soaked with 1 mM IPTG (running from top to bottom in the middle). 10 µl of 100 µM CAI-1 in DMSO/water was spotted at the asterisk, the plate was tipped and the solution was allowed to move left. As a surface active compound it regularly spread over a large area. Note that the bacterial lawn at left was not induced. Picture was taken from the bottom of the Petri-dish (three independent experiments were carried out, each with at least three agar plates and different concentrations of CAI-1 and IPTG; controls with solvent were negative).

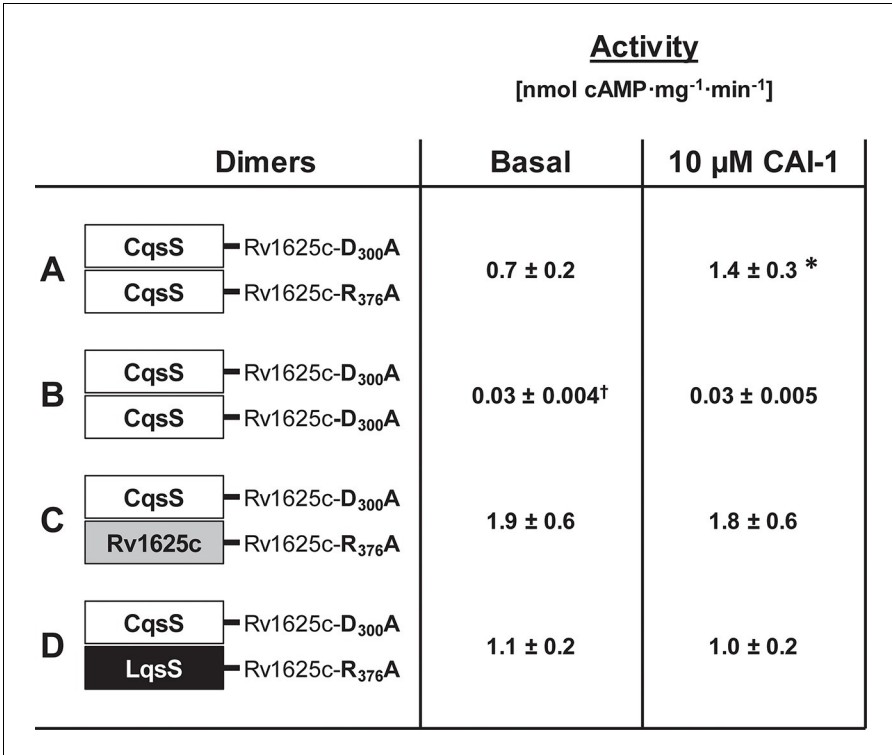

**Figure 5.** Homodimerization of the CqsS receptor is required for signaling. (**A**) with complementary Rv1625c point mutations Rv1626cD300A and Rv1625cR376A a regulated dimeric chimera was generated (*p<0.05 compared to respective basal activity). (**B**) as a control the construct CqsS-Rv1625cD300A was expressed alone. It was inactive. (**C, D**) complementing mutants with differing membrane domains were active, yet unregulated. Basal activity of construct B significantly differed from those in constructs **A**, **C**, and **D** (†p<0.001).

AC activity and regulation by CAI-1 was observed (*Figure 5A*). This demonstrated that the membrane anchors of CqsS dimerize and allow individually inactive AC domains to form an active, ligand-regulated dimer. Expectedly, the expression product of CqsS-Rv1625cD300A was inactive (*Figure 5B*). The experiment did not answer the question whether ligand binding requires a CqsS dimer. This was addressed by linking the inactive Rv1625c monomers to the 6TM anchors of either Rv1625c or the *Legionella* LqsS QS-receptor. First, CqsS-Rv1625cD300A and the full-length Rv1625cR376A were jointly expressed in pETDuet-3. AC activity was observed, yet CAI-1 did not regulate (*Figure 5C*). Second, the QS-receptor from LqsS was joined with the inactive Rv1625cR376A in a similar manner. CqsS-Rv1625cD300A and LqsS-Rv1625cR376A were concomitantly expressed. As before, AC activity was restored, yet regulation by CAI-1 was absent (*Figure 5D*). This indicated that the membrane anchors were close enough to enable productive heterodimerization; however, a regulatory ligand-binding site was absent. Possibly one ligand molecule binds at the interface of a receptor homodimer as is the case in the chemotaxis receptors Tsr or Tar or $Ni^{2+}$-binding in PhoQ (*Cheung et al., 2008*; *Gardina and Manson, 1996*; *Kanchan et al., 2010*; *Mowbray and Koshland, 1990*; *Yang et al., 1993*). Because the highly lipophilic ligand does not allow meaningful receptor binding studies, presently this cannot be examined any further.

## Stimulation of AC Rv1625c activity by CAI-1 is irreversible

The QS-system of *Vibrio* controls virulence. At high cell density, CAI-1 is produced, released and binds to the extracellular CqsS receptor (*Wei et al., 2012*). In a multistep intracellular process this results in reduced production of virulence factors and allows the pathogen to escape from the host, thus spreading disease (*Higgins et al., 2007*; *Rutherford and Bassler, 2012*). Because extracellular loops for ligand binding are absent in CqsS and because of the lipophilicity of the ligand, CAI-1 may bind within the membrane segments of the receptor dimer. Therefore, the question whether CAI-1

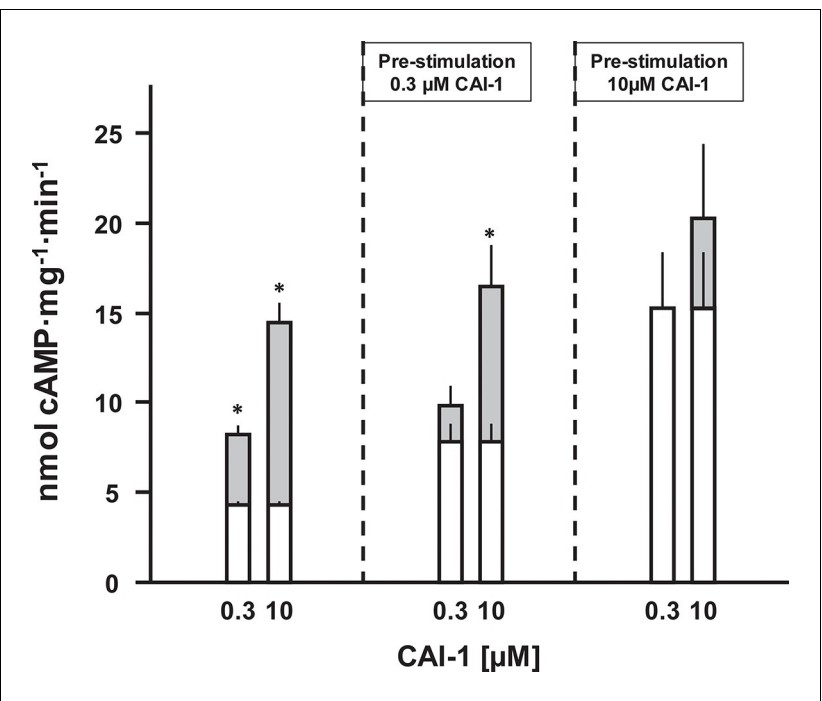

**Figure 6.** CAI-1 ligand binding to the CqsS QS-receptor is irreversible. Membranes containing CqsS$_{1-181}$F166L-Rv1625c$_{218-443}$ were stimulated with 0.3 or 10 µM CAI-1 (left), re-isolated and re-stimulated with 0.3 and 10 µM CAI-1. Only the stimulations marked with an asterisk differed significantly from the respective unstimulated controls. White bars represent basal AC activities, gray bars on top represent additive CAI-1-stimulated activities (S.E.M., n = 6).

stimulation is reversible was examined next. Membranes were stimulated with sub-saturating (300 nM) and saturating (10 µM) concentrations of CAI-1 for 10 min. The reactions were rapidly stopped by cooling to 0°C and the pre-stimulated membranes were re-isolated by ultracentrifugation. Those membranes were then stimulated again by CAI-1 (*Figure 6*). Membranes pre-treated with 300 nM CAI-1 had an elevated 'basal' AC activity equivalent to the previous 0.3 µM CAI-1 stimulation (*Figure 6*, left). Accordingly, re-stimulation by 0.3 µM CAI-1 failed whereas addition of 10 µM CAI-1 activated to the maximal possible extent. Membranes which were initially exposed to 10 µM CAI-1 remained almost fully activated and were completely refractory to re-stimulation (*Figure 6*). In this context we checked whether ligand was specifically binding at the receptor sites or remained unspecifically associated with the membranes. The supernatants of the ultracentrifugation steps of the pre-incubated membranes stimulated naïve membranes according to the previously tested CAI-1 concentrations. This excluded unspecific association of the lipophilic ligand with the membrane or adherence to the reaction vessels. Therefore, we can conclude that CAI-1 stimulation was irreversible. This is slightly reminiscent on the biochemistry of rhodopsin in the mammalian eye. There, retinal is even covalently bound into the membrane-segments of opsin, a GPCR with scant extra-membrane loops. After excitation receptor regeneration requires ligand removal by an enzymatic process and transport to the retinal pigment epithelium which, notably, contains exclusively adenylate cyclase type VII (*Völkel et al., 1996*). How in *Vibrio* signal termination is accomplished remains to be investigated. Possibilities are an inactivating metabolism of CAI-1 or proteolysis.

## 6TM membrane anchors in ACs, universal receptor modules?

The above data demonstrated that the homodimeric catalytic domain of the canonical class IIIa Rv1625c AC was capable to decode the ligand-initiated conformational signal of a 6TM QS-receptor and translate it into a change in AC activity. The functional membrane anchor replacement accompanied by a gain of a novel physiological function suggests that the Rv1625c 6TM anchor actually constitutes an orphan receptor, i.e. receptor without known ligand. Therefore, we examined the

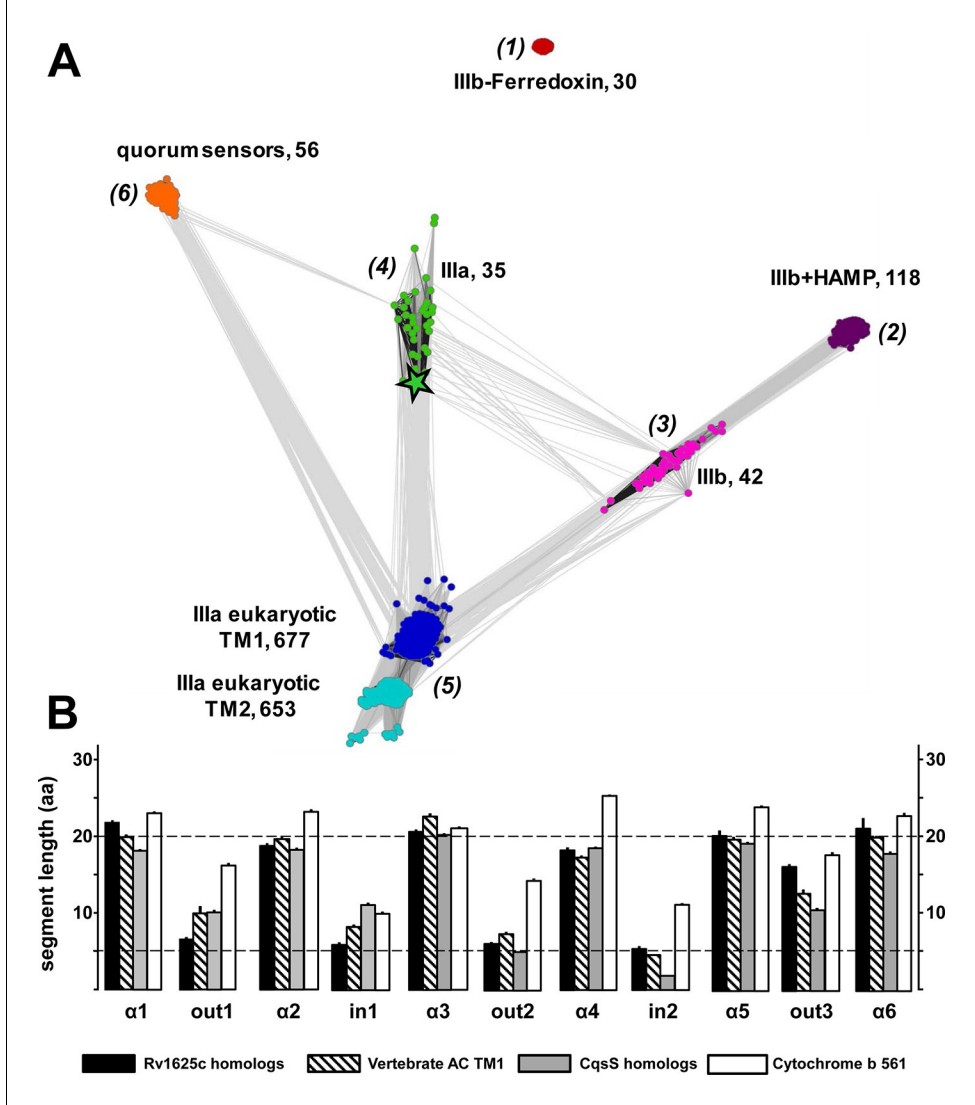

**Figure 7.** (A) Cluster map of 6TM domains of adenylate cyclases and CqsS-like sensory receptors. A comprehensive set of 6TM AC anchors was extracted from 408 eukaryotic and 1456 bacterial proteomes and clustered in CLANS (HHsearch p-value cutoff 5E-4, attraction value 10, repulsion value 5); outliers were removed. Each dot represents a single 6TM domain. Above threshold HHsearch hits are shown as connecting lines between AC pairs in different clusters; the darker line color, the more similar the protein sequences. 6TM anchors of ACs form five clusters of high pairwise sequence similarity: cluster (*1*), anchors of bacterial class IIIb ACs characterized by the presence of a cytosolic ferredoxin domain (30 sequences from the $\alpha$ and $\beta$ branches of proteobacteria). Cluster (*2*), anchors of bacterial class IIIb ACs characterized by a signal-transducing HAMP domain (118 sequences mainly from Actinobacteria, but also from $\alpha$-proteobacteria, $\delta$-proteobacteria, Chlorobia and Thermoleophilia). Cluster (*3*), anchors of bacterial class IIIb ACs similar to HAMP-associated anchor domains but which lack a HAMP domain (42 sequences mainly from $\alpha$-proteobacteria). Cluster (*4*), anchors from bacterial class IIIa ACs prototypically represented by the mycobacterial AC Rv1625c (35 from many different phyla of bacteria, including Actinobacteria, Proteobacteria, Chlorophyta, Spirochaetes, and Bacteriodetes). The enlarged asterisk denotes the position of the mycobacterial AC Rv1625c. Cluster (*5*), anchors of the pseudoheterodimeric eukaryotic class IIIa ACs (TM1 677 sequences, TM2 653 sequences). CqsS Cluster (*6*), 6TM domains of sensory His-Kinases similar to CqsS (from Bacteriodetes, Chlorobia, $\alpha$-, $\beta$-, and $\gamma$-proteobacteria). (B) Length comparisons of the transmembrane helices and loops of 6TM membrane anchors/receptors/sensors. The data sets from clusters 4, 5 and 6 from *Figure 7A* were used supplemented with 250 cytochrome b561 proteins. In case no S.E.M. is visible the size of the vertical bar is within the line thickness of the respective bar. '$\alpha$'-Numbering denotes the consecutive TM helices starting from the N-terminus, 'out' and 'in' denote sequential extra- and intracellular loop sequences. The two horizontal lines are at the 5 and 20 aa level.

*Figure 7 continued on next page*

*Figure 7 continued*

The following figure supplements are available for figure 7:

**Figure supplement 1.** Length comparisons of α-helices and loops of 6TM modules from adenylate cyclases and quorum sensors.

**Figure supplement 2.** The 12 sequences used for the alignment were from *Agrobacterium albertimagni*, WP_006725538; *Arthrospira maxima*, B5VUZ0; *Arthrospira platensis*, D5A5G2; *Beggiatoa*, A7BXS6; *Dechloromonas*, Q47AI8; *Hyphomicrobium*, C6QBG1; *Lyngbya*, A0YQ82; *Mesorhizobium*, LSHC420B00; *Microcoleus sp.*, PCC_7113; *Oscillatoria acuminata*, PCC_6304; *Nostoc*, YP_001866931; *Mycobacterium tuberculosis*, ALB18789 (Rv1625c).

**Figure supplement 3.** AC_1 Sequences used for the alignment: *Macaca fasci*, XP_005551359.1; *Bos taurus*, NP_776654.1; *Anas platyrhynchos*, XP_005014606.1; *Danio rerio*, NP_001161822.1; *Mesocricetus auratus*, XP_005083033.1; *Pseudopodoces humilis*, XP_005525280.1; *Gallus gallus*, XP_418883.4; *Homo sapiens*, NP_066939.1; *Mouse*, GI:62512159; *Heterocephalus glaber*, XP_004840600.1; *Orcinus orca*, XP_004283063.1; *Pan troglodytes*, XP_519081.3; *Ficedula albicollis*, XP_005041477.1; *Sarcophilus harrisii*, XP_003762599.1; *Odobenus rosmarus divergens*, XP_004409496.1; *Melopsittacus undulatus*, XP_005148091.1.

---

relationship between the QS-receptors and the 6TM AC membrane anchors by a focused bioinformatic approach.

From the Uniprot Reference Proteomes databank we obtained all class III ACs using the highly conserved catalytic domain for data mining (see Materials and methods). The catalytic domains were stripped; the putative 6TM domains were extracted and retained. Similarly, we extracted 6TM modules related to the CqsS quorum-sensing receptor. The combined data set comprising a total of 1616 6TM modules was subjected to a cluster analysis using the CLANS software (*Figure 7A*; *Frickey and Lupas, 2004*). In total, six distinct clusters were observed, five of which were interconnected to different extents, among them the cluster of QS-receptors (*Figure 7A*). This clear-cut separation of 6TM modules is surprising and revealing. Transmembrane spanning α-helices have a strong predilection for hydrophobic residues with Leu, Ile, Val, Ala, Gly and Phe comprising two-thirds (*Senes et al., 2000*). Yet, the unequivocal clustering indicates a differentiated pattern of conserved 6TM variants which is at odds with an assignment of simply an anchoring device. We propose that the diversity of conserved patterned subtypes mirrors a succinct adaptation to particular physiological functions. By far the majority of 6TM modules were eukaryotic ACs, from the slime mold *Dictyostelium*, the extant coelacanth up to man. The preponderance of eukaryotic sequences is, at least in part, due to the fact that in vertebrates nine distinct membrane-delimited pseudoheterodimeric AC isoforms exist, each with two TM modules, TM1 and TM2. The observed clusters were identified as follows: (*1*) An isolated, unconnected cluster of 30 bacterial class IIIb ACs is characterized by a ferredoxin between the membrane and the catalytic domain. A representative example is an AC from *Rhodopseudomonas palustris* (Uniprot Q132R4) which intracellularly appends to a ferredoxin module. The membrane anchors show sequence similarity to the cytochrome b subunits present in the fumarate reductase/succinate dehydrogenase protein families to an extent that they can be matched by simple BLAST searches. These 6TM modules have four precisely spaced intra-membrane histidine residues which coordinate heme as an electron carrier (*Einsle et al., 2000*; *Hederstedt, 1998*; *Kern et al., 2010a*; *2010b*). It can confidently be predicted that the AC 6TM anchors of this cluster will turn out to contain two heme entities as prosthetic groups. (*2*) A cluster of 118 bacterial class IIIb ACs is characterized by a signal-transducing HAMP domain between membrane anchor and catalytic domain such as the well-studied mycobacterial AC Rv3645 (*Hazelbauer et al., 2008*; *Hulko et al., 2006*; *Kanchan et al., 2010*; *Mondéjar et al., 2012*). (*3*) A cluster of 42 bacterial class IIIb ACs shows partly high local similarity to ACs of cluster (*2*) as visualized by the number and intensity of the gray colored connecting lines between both clusters (*Figure 7A*). Yet, ACs in cluster (*3*) lack a HAMP domain. To our knowledge, none of these ACs has ever been interrogated experimentally. (*4*) A cluster of 35 membrane anchors from bacterial class IIIa ACs as prototypically represented by the mycobacterial AC Rv1625c used here. It is predominantly connected to the cluster with mammalian TM modules. (*5*) Two tightly connected groups are combined here into one cluster of 677 TM1

and 653 TM2 modules which are derived from eukaryotic, mostly vertebrate, pseudoheterodimeric class IIIa ACs. The difference in the number of TM1 and TM2 units is due to individual sequence divergences of the TM2 domains, database miss-annotations, and problems in the automated prediction of transmembrane spans. (6) A cluster of 56 6TM modules corresponds to quorum-sensing receptors of the CqsS type.

The cluster of the eukaryotic TM1 domain (from cluster 5) was connected to the clusters of the bacterial type IIIa (4) and type IIIb (3) by a large number of pairwise matches, typically covering all six transmembrane spans. In contrast, the CqsS cluster had significantly fewer and only local matches, aligning quorum-sensing sequences specifically to transmembrane spans 1 and 2 of individual eukaryotic AC TM1 sequences (all from the bony fish Osteichtyes), and to TM 3, 4 and 5 of the bacterial type IIIa ACs, each at approximately 20–25% identity. Notably, the average sequence identity within the clusters themselves is approximately 40%. Thus, currently we can only speculate whether the observed similarities indicate remote homology at the limit of recognition or convergence due to identical structural and functional constraints.

Visual inspection of sequence properties of single TMs indicated a remarkable shortness of $\alpha$-helices and respective interconnecting loops already apparent in the 2D presentations (*Figure 1*). Notably, the loops are by far shorter than those of comparable sensory proteins, such as mammalian GPCRs with seven TMs, bacterial chemotaxis receptors, His-kinases and ACs with 2 TMs or 4 TMs such as the chase domain (**c**yclase/**h**istidine kinases **a**ssociated **s**ensory **e**xtracellular). To support this notion we investigated the length parameters of the transmembrane segments and loops of AC and CqsS anchor types which were used in the cluster analysis, and compared them to 250 orthologues of cytochrome b561 (*Figure 7B* and *Figure 7—figure supplement 1*). The latter protein was selected because a high-resolution structure is available, i.e. they form a 2x6TM homodimer of short TM spans and loops analogous to those from Rv1625c and CqsS (*Lu et al., 2014*). The lengths of the six TM segments and of the five loops were well conserved between orthologues of the same anchor type and highly similar between anchors of the bacterial AC classes IIIa and IIIb, eukaryotic TM1, and, most surprisingly, CqsS-like QS-receptors (*Figure 7B*). On the other hand, the eukaryotic TM2 anchors, those of the ferredoxin-associated class IIIb, and anchors coupled to a cytosolic HAMP-domain (class IIIb ACs in cluster 2) had one or even several elongated loops or $\alpha$-helices (*Figure 7* and *Figure 7—figure supplement 1*).

The short helices of around 20 residues in this 6TM module design must cross the 30 Å thick lipid bilayer almost orthogonally and the short connecting loops restrict the number of positional permutations foreshadowing a compact packing. Therefore, the structures of transmembrane domains of eukaryotic AC TM1, the bacterial class IIIa ACs such as Rv1625c, and QS-receptors of the CqsS type can be reasoned to possess overall structural similarities with that of cytochrome b561.

## Discussion

In 1989 the analysis of the first aa sequence of a mammalian AC suggested that the two membrane anchors, each consisting of six transmembrane segments, might carry a transporter or channel function (*Krupinski et al., 1989*). Since then, our knowledge concerning the membrane anchors has not advanced; to date no function for approximately 40% of a class IIIa AC protein sequence has been identified which goes beyond a mere membrane fixation. After 26 years we demonstrate for the first time that a canonical class IIIa AC with a 6TM membrane anchor is directly regulated by a membrane receptor of an identical design, yet with a known ligand, the QS-receptor from *V. harveyi*.

Our data raise novel questions concerning the evolution of the regulation of class III ACs with 6TM membrane-anchoring modules. In absence of bacterial G-proteins bacterial class III ACs probably will turn out to be directly regulated via their prominent membrane anchors with the above mentioned *R. palustris* AC as an emerging example. In fact the membrane anchors of the class IIIa bacterial ACs are highly diverged whereas the catalytic domains are conserved (*Figure 7* and *Figure 7—figure supplement 2*). This suggests that in bacteria the TM domains have evolved independently and very rapidly relative to the catalytic domains. This is in agreement with the general observation that mutations in upstream regulatory domains mostly are neutral and over time a variety of ligand specificities have evolved by chance mutations. At the same time even slightly detrimental mutations in downstream effector domains are not tolerated, thus resulting in sequence and functional conservation (*Schultz and Natarajan, 2013*). In such an evolutionary scenario the

mechanisms of signal transduction per se remain intact and allow revealing combinatorial diversity as demonstrated here (*Schultz et al., 2015*).

In contrast, in vertebrates the membrane-delimited ACs are regulated indirectly by GPCR activation which intracellularly results in release of Gα proteins. Is this the result of a loss of direct ligand regulation during evolution while indirect GPCR regulation evolved or has ligand regulation of mammalian ACs been missed so far? Because the AC membrane bundles of vertebrates are highly conserved in an isoform-specific manner from the coelacanth to man (*Figure 7* and *Figure 7—figure supplement 3* as an example of mammalian AC_I and AC_II subtypes) one can reasonably assume that they have an indispensable physiological function. This could be a particular compartmentalized membrane localization e.g. described in (*Crossthwaite et al., 2005*) or, in our view more likely and in accordance with our findings an as yet hidden receptor function, as a direct ligand-binding module or in conjunction with an accessory membrane protein that operates as the true sensor. The second alternative would suggest that in vertebrates regulation of intracellular cAMP concentrations is subject to an interaction between direct, ligand-mediated and indirect GPCR-Gsα-regulated effects. A lasting receptor occupation by a ligand during different physiological states might well set the responsiveness of mammalian AC isoforms to a transient GPCR-Gsα activation. This would allow that lasting and transient physiological conditions converge via direct and indirect regulation in a central second messenger system, a situation absent in bacteria. This concept poses the questions whether in vertebrates suitable extracellular signals exist and whether the molecular provisions for a direct signal transduction through the 9 AC isoforms have been evolutionarily conserved. We are currently exploring these questions.

Recently we demonstrated that we could functionally replace the bacterial AC 6TM anchors by the *E. coli* chemotaxis receptors for serine or aspartate (*Kanchan et al., 2010*; *Schultz et al., 2015*; *Winkler et al., 2012*). By analogy, the results supported the hypothesis of a receptor function for the AC membrane anchors. A less plausible interpretation would have been that the functional coupling might just be a manifestation of the modular composition of signaling proteins. The compatibility may only have required that a satisfactory domain order is maintained in such chimeras without invoking a functional relatedness. The data reported here add an entirely novel dimension to the working hypothesis that AC membrane anchors in bacterial homodimeric as well as in mammalian pseudoheterodimeric ACs function as ligand receptors. The 2D models of the membrane anchors of Rv1625c and CqsS are similar (see *Figure 1*). This should allow an almost isosteric replacement. Competent QS-receptor and cyclase chimeras were dependent on the point of linkage indicating that the connecting sites had been evolutionarily predesigned, were conserved and fully operational for signal transduction between functionally differing proteins (*Schultz and Natarajan, 2013*). Implicitly this supports the prediction that the 6TM anchors of such ACs have a receptor function for which stimuli have yet to be identified. Because the ligands for CqsS and LqsS are known, the direct regulation of AC activity then is no real surprise. Admittedly, based alone on the biochemical data one might again take pains to argue that the exchangeability of 6TM-anchors just extends the range of signaling modules which can structurally and possibly functionally replace each other. This alternative interpretation is rather implausible in our view. First mixing and matching TM domains of eukaryotic AC is impossible without loss of activity (*Seebacher et al., 2001*). Second, the cluster analysis visualizes similarities between individual pairs of CqsS receptor type modules and membrane anchors from class IIIa ACs. Thus it supports the hypothesis that the membrane anchors of class III ACs, bacterial and mammalian alike have a function beyond membrane fixation. In this context cluster (*1*), which is unrelated to all other clusters, is particularly interesting. It demonstrates that comparable transmembrane architectures can result in significantly different sequence patterns notwithstanding the similar amino acid composition commonly shared by all membrane domains. Currently, it is impossible to speculate about or even predict the nature of ligands for the 6TM AC modules examined here. One might expect, however, that they will be closely associated with the intra-membrane space as extra-membranous loops for ligand-binding are noticeably absent in this type of 6TM bundles.

## Materials and methods

An *E.coli* culture containing the protein CqsS of *V. harveyi* was obtained from K. Jung and LqsS from H. Hilbi, LMU, Munich (Germany). *M. tuberculosis* AC Rv1625c and point mutants D300A and R376A

were available in the laboratory (*Guo et al., 2001*). Radiochemicals were from Hartmann Analytik (Braunschweig, Germany) and Perkin Elmer (Rodgau, Germany). Enzymes were from either New England Biolabs or Roche Diagnostics. Other chemicals were from Sigma, Roche Diagnostics, Merck and Roth. CAI-1, LAI-1 and 3,4-tridecanediol were synthesized in-house according to (*Bolitho et al., 2011*; *Ng et al., 2010*). Concentration-response curves were usually limited to maximally 10 µM as the ligands have surfactant properties and assays with higher concentrations tended to give inconsistent results.

## Plasmid construction

In CqsS-Rv1625c chimeras the following CqsS receptor length variants were probed in different combinations: F168; V172; K177; A181; S183; G185; G187; I188; H190; P195;L196. For AC Rv1625c catalytic domains the tested length variants were: A201, L202, R203 and R218. Standard molecular biology methods were used for DNA manipulations (primer sequences are in *Supplementary file 1*). DNA fragments and vectors were restricted at their 5′BamHI or EcoRI and 3′HindIII sites and inserted into pQE80$_L$ ($\Delta$ XhoI; $\Delta$ NcoI). When appropriate, silent restriction sites were introduced. All constructs carried an N-terminal His$_6$-tag for detection in Western blots. In the pETDuet-3 vector the first MCS was been replaced by that of pQE30 introducing an N-terminal His$_6$-tag. The second MCS in pETDuet-3 carried a C-terminal S-tag for Western blotting. The fidelity of all constructs was confirmed by double-stranded DNA sequencing.

## Protein expression

Constructs were transformed into *E. coli* BL21(DE3). Strains were grown overnight in LB medium (20g LB broth/l) at 37°C containing 100 µg/ml ampicillin. 200 ml LB medium (with antibiotic) was inoculated with 5 ml of a preculture and grown at 37°C. At an A$_{600}$ of 0.3, the temperature was lowered to 22°C and the expression was started with 500 µM isopropyl thio-$\beta$-D-galactoside (IPTG) for 2.5–5 hrs. Cells were harvested by centrifugation, washed once with 50 mM Tris/HCl, 1mM EDTA, pH 8 and stored at -80°C. For preparation of cell membranes cells were suspended in lysis buffer (50 mM Tris/HCl, 2 mM 3-thioglycerol, 50 mM NaCl, pH 8) containing complete protease inhibitor cocktail (Roche Molecular, Mannheim, Germany) and disintegrated by a French press (1100 p.s.i.). After removal of cell debris (4.300 x g, 30min, 4°C) membranes were collected at 100000 x g (1h at 4°C). Membranes were suspended in buffer (40 mM Tris/HCl, pH8, 1.6 mM 3-thioglycerol, 20% glycerol) and assayed for AC activity. A more detailed description of protein expression and membrane preparation is available as described in detail at Bio-protocol (*Beltz and Schultz, 2016*).

## Adenylate cyclase assay

Adenylyl cyclase activity was determined for 10 min in 100 µl at 37°C (*Salomon et al., 1974*). The reactions contained 5 µg protein, 50 mM Tris/HCl pH 8, 22% glycerol, 3 mM MnCl$_2$, 6 mM creatine phosphate and 230 µg creatine kinase, 75 µM [$\alpha$-$^{32}$P]-ATP, and 2 mM [2,8-$^3$H]-cAMP to monitor yield during cAMP purification. Substrate conversion was kept below 10%.

## Western blot analysis

The integrity of expressed recombinant membrane proteins was probed by Western blotting. Sample buffer was added to the membrane fractions and applied to SDS-PAGE (12 or 15%), in which proteins were separated according to size. For Western blot analysis, proteins were blotted onto PVPF membrane and examined with an RGS-His$_4$-antibody (Qiagen, Hilden, Germany ) or S-tag antibody (Novagen R&D systems, Darmstadt, Germany) and a 1:2500 dilution of the fluorophore conjugated secondary antibody Cy3 (ECL Plex goat-$\alpha$-mouse IgG-Cy3, GE Healthcare, Freiburg, Germany). Detection was carried out with the Ettan DIGE Imager (GE Healthcare). In general, proteolysis of expressed proteins was not observed.

## Bioinformatics

### Dataset

Sequences were taken from the Uniprot Reference Proteomes databank (release 2015_04). AC anchor sequences were identified by their conserved catalytic domain in a HMMer3 search (E-value cutoff 1E-5; (*Eddy, 2011*) with EBI SMART's cycc family alignment (*Schultz et al., 1998*). To extract

putative membrane domains, all segments N-terminal of cycc hits, up to the protein N-terminus or another cycc domain, were extracted. HAMP domains, if present, were located using HMMer3 and the HAMP EBI SMART family alignment and then removed. The extracted sequences were clustered to 30% sequence identity using kClust (*Hauser et al., 2013*). Then, the clusters were aligned individually and their TM spans were predicted by Polyphobius (*Kall et al., 2005*) and manual inspection. 6TM clusters were merged and used for further analyses. CqsS and cytochrome b561 homologs were identified in HMMer3 jackhmmer searches, using the *V. cholerae* CqsS and *A. thaliana* Cy561 anchors (Uniprot identifiers Q9KM66 and Q9SWS1) as queries.

## Cluster analysis

For every sequence in the 6TM AC anchor data set we searched the Uniprot database, clustered at 20% sequence identity (Uniprot20, June 2015, available at http://toolkit.tuebingen.mpg.de), with HHblits (p-value cutoff 1E-3, minimum coverage 50%, two iterations; (*Remmert et al., 2012*). The resulting sequence alignments were used to create profile hidden Markov models (HMMs) that included Polyphobius TM predictions instead of secondary structure annotation. The same procedure was applied to CqsS anchor homologs. The clustering was performed using the CLANS software (*Frickey and Lupas, 2004*), based on pairwise HMM-HMM comparisons (*Soding, 2005*).

## Helix/loop length analysis

The helix and loop lengths were measured using the sequences taken from the respective clusters of the CLANS analysis and cytochrome b561 homologs. Predicted transmembrane spans were based on manually refined Polyphobius predictions.

## Statistical analysis

All experiments were repeated at least thrice. Data are presented as means ± S.E.M. when applicable. Student's t test was used.

## Acknowledgements

This publication is dedicated to Prof. Dr. Günter Schultz, FU Berlin, at the occasion of his 80th birthday. Supported by Deutsche Forschungsgemeinschaft (SFB 766; TP B08) and institutional funds of the Max-Planck-Society. We are grateful to Prof. Dr. K. Hantke who carried out the *in vivo* experiments and to Prof. Dr. P. Koch for the synthesis of 3, 4-tridecanediol. We thank A. Schultz for invaluable technical assistance. We are indebted to Prof. Dr. A. Lupas for continuous encouragement and critical discussions.

## Additional information

### Funding

| Funder | Grant reference number | Author |
| --- | --- | --- |
| Deutsche Forschungsgemeinschaft | SFB 766 | Stephanie Beltz |
| Max-Planck-Gesellschaft | | Jens Bassler |

The funders had no role in study design, data collection and interpretation, or the decision to submit the work for publication.

### Author contributions

SB, JB, Acquisition of data, Analysis and interpretation of data, Drafting or revising the article; JES, Conception and design, Acquisition of data, Analysis and interpretation of data, Drafting or revising the article

### Author ORCIDs

Joachim E Schultz, http://orcid.org/0000-0002-1985-4853

## Additional files

**Supplementary files**
• Supplementary file 1. List of sense (s) and antisense (as) primers used for generating the diversity of chimeras used in this study.

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
