## [Decision Letter]

Thank you for submitting your work entitled "The membrane anchors of adenylate cyclases: regulation by the quorum sensor from Vibrio indicates a receptor function" for consideration by *eLife*. Your article has been reviewed by two peer reviewers including Stephen Sprang, and the evaluation has been overseen by Michael Marletta as the Senior Editor and Reviewing Editor.

The reviewers have discussed the reviews with one another and the Reviewing Editor has drafted this decision to help you prepare a revised submission.

From the protein-engineering standpoint, your paper is an interesting and important contribution that should be of interest to a broad range of scientists who work in the area of signal transduction at the molecular level.

There are some points of concern that need to addressed. The data shown in Figure 3 do not allow calculation of the maximum level of activation. Perhaps it is not possible to achieve ligand concentrations above 10 µM. Hence the ~6-fold activation of the CqsS_1-181_F166L-Rv1625c_218-443_ chimera by CAI-1 may be an underestimate. Even so, this is considerably lower than the 10^[5]^ to 10^[4]^-fold activation of CqsS-stimulated reporter gene transcription observed at 0.01 µM ligand in *V. harveyi* and *V. cholerae* (Ng et al., 2011). More than ligand-induced dimerization may be involved in the coupling between the CqsS TM/ligand-binding domain and its authentic effector, including allosteric and orientation factors, and they are clearly not retained when the TM domain is linked to an exogenous effector domain. Also, is anything known about the ability of CqsS to discriminate between the ligands used here? These points need to be discussed.

Figure 4 is difficult to interpret. The IPTG strip was placed down the center of the dish, but the light/dark boundary is off to the left. Is the ligand only applied at the leftward vertical boundary? This is an experiment in which n = 1. Have you reproduced this experiment?

Figure 7 indicates at least one connection between the quorum sensor domains and IIIa AC, and many connections with the IIIa eukaryotic AC TM1s. We assume that this means that there is actual sequence identity between them. This would be a strong case for divergence of CsqS regulatory and AC TM domains. The authors should address this either with a figure or by citing ranges of% amino acid sequence identities. If regions of identity are clustered, for example, near the effector-TM linker region, this would be particularly interesting.

Abstract (first sentence): We do not agree that all ACs convert extracellular stimuli. In particular in bacterial AC, there are often sensor domains for intracellular ligands. It can be a philosophical question, in the end almost every intracellular signal can be traced back to some extracellular one, but in a more narrow sense, some ACs do respond to intracellular signals.

Abstract and in the text: The expression "indirect regulation" when referring to GPCR/G-proteins is not the best terminology. It would be more precise to say "indirect regulation by GPCRs", since regulation by G-proteins is direct. We think you should consider rephrasing this. (See also the first paragraph of the Introduction).

---

## [Author Response]

*There are some points of concern that need to addressed. The data shown in Figure 3 do not allow calculation of the maximum level of activation. Perhaps it is not possible to achieve ligand concentrations above 10 µM. Hence the ~6-fold activation of the CqsS_1-181_F166L-Rv1625c_218-443_ chimera by CAI-1 may be an underestimate. Even so, this is considerably lower than the 10^[5]^ to 10^[4]^-fold activation of CqsS-stimulated reporter gene transcription observed at 0.01 µM ligand in V. harveyi and V. cholerae (Ng et al., 2011). More than ligand-induced dimerization may be involved in the coupling between the CqsS TM/ligand-binding domain and its authentic effector, including allosteric and orientation factors, and they are clearly not retained when the TM domain is linked to an exogenous effector domain.*

This is correct and the recommended points of discussion have been incorporated, partly using the phrasing suggested.

*Also, is anything known about the ability of CqsS to discriminate between the ligands used here? These points need to be discussed.*

Meanwhile we obtained a ligand-like compound (3,4-dihydroxy-tridecane). The corresponding data have been added in Figure 3 and the figure legend has been changed accordingly. We regret that no other related chemicals are available to us.

*Figure 4 is difficult to interpret. The IPTG strip was placed down the center of the dish, but the light/dark boundary is off to the left. Is the ligand only applied at the leftward vertical boundary? This is an experiment in which n = 1. Have you reproduced this experiment?*

Correct point. Of course this experiment has been reproduced several times. We show the same plate with a different, more contrasting brightness to better visualize the bacterial lawn in the un-induced area. As the ligand has surfactant-like properties exact handling is included in the legend.

*Figure 7 indicates at least one connection between the quorum sensor domains and IIIa AC, and many connections with the IIIa eukaryotic AC TM1s. We assume that this means that there is actual sequence identity between them. This would be a strong case for divergence of CsqS regulatory and AC TM domains. The authors should address this either with a figure or by citing ranges of% amino acid sequence identities. If regions of identity are clustered, for example, near the effector-TM linker region, this would be particularly interesting.*

The lines indicate above-threshold scoring HMM-HMM alignments. This is based on similar position-specific amino acid compositions and may not always imply actual identity between two sequences. Nonetheless, we agree that the section needs a more to the point description of the actual matches. We have added a short paragraph to this section, so that it now explicitly states the aligning ranges and approximate identities (unfortunately this is difficult to read in bioinformatic speak). The data are not suitable for a graphic item.

*Abstract (first sentence): We do not agree that all ACs convert extracellular stimuli. In particular in bacterial AC, there are often sensor domains for intracellular ligands. It can be a philosophical question, in the end almost every intracellular signal can be traced back to some extracellular one, but in a more narrow sense, some ACs do respond to intracellular signals.*

Correct. Has been rephrased to include other possibilities.

Abstract and in the text: The expression "indirect regulation" when referring to GPCR/G-proteins is not the best terminology. It would be more precise to say "indirect regulation by GPCRs", since regulation by G-proteins is direct. We think you should consider rephrasing this. (See also the first paragraph of the Introduction).

Correct. This has been changed throughout the manuscript by rephrasing.